# Analysis of Tumor Microenvironment Heterogeneity among Breast Cancer Subtypes to Identify Subtype-Specific Signatures

**DOI:** 10.3390/genes14010044

**Published:** 2022-12-23

**Authors:** Ji Li, Jiashuo Wu, Junwei Han

**Affiliations:** 1College of Bioinformatics Science and Technology, Harbin Medical University, Harbin 150086, China; 2Bio-Pharmaceutical Key Laboratory of Heilongjiang Province, Harbin Medical University, Harbin 150086, China

**Keywords:** breast cancer subtypes, tumor microenvironment, immune hot and cold tumors, prognostic biomarkers, subtype-specific treatment approaches

## Abstract

Breast cancer is one of the most frequent malignancies in women worldwide. According to 50-gene signature, Prediction Analysis of Microarray 50 (PAM50), breast cancer can be categorized into five molecular subtypes, and these subtypes are highly heterogeneous in different molecular characteristics. However, the landscape of their tumor microenvironment (TME) heterogeneity has not been fully researched. Using the multi-omics dataset of breast cancer from the METABRIC cohort (*n* = 1699), we conducted extensive analyses of TME-related features to investigate TME heterogeneity in each breast cancer subtype. We then developed a cell-based subtype set enrichment analysis to identify the subtype-specific TME cells, and further evaluate their prognostic effects. Our results illustrate that different breast cancer subtypes exhibit different TME patterns. Basal-like and HER2-enriched subtypes are associated with high immune scores, expression of most immune regulatory targets, and immune cell infiltration, suggesting that these subtypes could be defined as “immune hot” tumors and suitable for immune checkpoint blockade (ICB) therapy. In contrast, Luminal A and Luminal B subtypes are associated with low immune scores and immune cell infiltration, suggesting that these subtypes could be defined as “immune cold” tumors. Additionally, the Normal-like subtype has relatively high levels of both immune and stromal features, which indicates that the Normal-like subtype may be suitable for more diverse treatment strategies. Our study reveals the breast cancer tumor microenvironment heterogeneity across subtypes. The comprehensive analysis of breast cancer TME-related characteristics may help us to adopt a tailored treatment strategy for different subtypes of patients.

## 1. Introduction

Breast cancer is one of the most frequent malignancies in women worldwide; it is a molecularly and histologically diverse illness with at least five molecular subtypes. Based on the expression levels of 50 genes (PAM50), breast cancer can be mainly classified into Luminal A, Luminal B, HER2-enriched, Basal-like, and Normal-like subtypes [1]. The patients with different subtypes are generally diverse in mutational profiles, expression profiles, and biological processes and pathways, which may result in differences in progression, response to immunotherapy, and prognosis outcome of cancer. Thus, different subtypes should be treated with different therapeutic strategies to implement precise treatment.

In the last few years, the tumor microenvironment (TME) has long been recognized as a key factor that affects cancer initiation, development, the response to immunotherapy, and clinical outcome in breast cancer [2,3]. The TME refers to tumor cells, immune cells (B cells, T cells, macrophages, dendritic cells, natural killer cells, etc.), stromal cells (fibroblasts, endothelial cells, pericytes, and myoepithelial cells, etc.), and other cells, of which stromal cells and immune cells are two main components in the TME, and they were reported to play important roles for diagnostic and prognostic evaluations of cancer [4]. Several studies have proposed that the profile and functionality of immune cell infiltration were diverse in different breast cancer subtypes [5,6,7]. For example, TNBC is a subtype of breast cancer with a unique tumor microenvironment. Factors such as immunosuppressive cells and epigenetic modifications in the microenvironment can combine to promote tumor growth as well as metastasis. These features make the microenvironment of TNBC a targeted therapeutic option [8]. Bareche et al. further classified TNBC and found that there remains a high degree of heterogeneity among the TME of TNBC molecular subtypes, as evidenced by specific immune or stromal infiltration and metabolic processes [9]. Therefore, understanding breast cancer subtype-specific TME features may help to gain insight into the TME contributes to the progression of cancer and develop the subtype-specific biomarkers used in immunotherapy for breast cancer patients.

The TME cells are frequently distinguished by cell-type-specific marker genes [4]. According to these marker genes, several computational methods were developed to infer the infiltrated abundance of immune cells in TME using gene expression profiles of bulk tumors. These methods can be mainly classified into two categories. The first mainly uses the deconvolution algorithm to estimate the TME cell infiltrated levels, such as CIBERSORT [10]. The second mainly uses the single-sample gene set enrichment analysis (ssGSEA) algorithm to estimate the relative abundance of TME cells by assigning curated gene sets to represent cell types, such as ImmuCellA [11]. xCell integrates the advantages of gene set enrichment with deconvolution approaches to enumerate the abundance of TME cells [12]. These methods provided a new way to dissect the cellular heterogeneity and understand the effect of TME cells in cancer subtypes.

Here, in order to investigate whether breast cancer subtypes have heterogeneous TME phenotypes, we performed a comprehensive analysis to investigate subtype-based differences in TME-related features, such as tumor mutation burden (TMB), immune score, stromal score, major histocompatibility complex (MHC) score, T-cell-inflamed gene expression profile (GEP), and the TME-related signature sets. We then developed a cell-based subtype set enrichment analysis to identify the subtype-specific infiltrate levels of TME cells, and further evaluated their prognostic effects in the corresponding subtypes.

## 2. Materials and Methods

### 2.1. Datasets

We first achieved a breast cancer dataset (METABRIC, Nature 2012 & Nat Commun 2016) [13,14,15] from the cBioPortal database (https://www.cbioportal.org/ (accessed on 1 May 2022)), which is composed of 2509 breast cancer samples. The gene expression profile, somatic mutation profiles, copy number variation (CNV) data, clinical information, and overall survival data were extracted for further analysis. After filtering samples with PAM50 subtype information, 1699 samples were included (Basal-like, N = 199; HER2 = enriched, N = 220; Luminal A, N = 679; Luminal B, N = 461; Normal-like, N = 140). Of the 1699 samples in METABRIC cohort, a total of 1672 samples have mutation information. We then downloaded the TCGA breast cancer cohort from the GDC TCGA data portal (https://portal.gdc.cancer.gov/ (accessed on 1 May 2022)), which includes 1217 breast cancer samples, and 956 samples have subtype information (Basal-like, N = 142; HER2, N = 67; Luminal A, N = 437; Luminal B, N = 192; Normal-like, N = 118), to validate our results. For gene expression data, FPKM-normalized profiles were used, and all expression values were then log2 (value + 1) transformed.

### 2.2. Analysis of TME-Related Gene Signatures

We first collected eight gene signature sets related to the TME, including two immune-related (“cytolytic activity” and “lymphocytes”) [16,17], two vascularization-related (“hypoxia” and “lymphangiogenesis”) [18,19], one stromal-related (“stromal”) [20], and three metabolism-related (“glycolysis”, “lipid metabolism”, and “pentose phosphate pathway”) signatures [21,22,23]. The single-sample gene set enrichment analysis (ssGSEA) method was used to calculate the enrichment scores of the TME-related signature sets through the GSVA Bioconductor package [24]. The association of each TME signature set with subtype was assessed by Mann–Whitney U test.

We then calculated immune score, stromal score, and tumor purity for each patient through the R package “estimate” based on gene expression data [25]. Next, we evaluated the major histocompatibility complex (MHC) score and T-cell-inflamed gene expression profile (GEP). The MHC score was formulated from the gene expression levels of the “core” MHC-I set (including HLA-A, HLA-B, HLA-C, TAP1, TAP2, NLRC5, PSMB9, PSMB8, and B2M). The FPKMs of these genes were first log-transformed and then median-centered across all patients in the dataset. The mean expression levels of these core genes were then calculated for each patient as the MHC score [26]. The T-cell-inflamed GEP was composed of 18 inflammatory genes that are related to antigen presentation, chemokine expression, cytolytic activity, and adaptive immune resistance, including CCL5, CD27, CD274 (PD-L1), CD276 (B7-H3), CD8A, CMKLR1, CXCL9, CXCR6, HLA-DQA1, HLA-DRB1, HLA-E, IDO1, LAG3, NKG7, PDCD1LG2 (PDL2), PSMB10, STAT1, and TIGIT [27]. The ssGSEA normalized enrichment score of these genes was calculated as the GEP score for each sample [28]. Finally, tumor mutation burden (TMB) was calculated as the total number of non-silent somatic mutations per coding area of genome for each sample in the study.

### 2.3. Calculation of Effect Size to Evaluate the Difference in TME-Related Signatures between Samples of Two Groups

For measuring the difference in TME-related signatures between samples of two groups, we chose Cohen’s d measure, which allowed us to calculate the effect size for different sample groups by taking the group size into account in the computation of the pooled standard deviation. Cohen’s d was calculated as follows:(1)$$\mathrm{cohen}’s\;d=\frac{M_{1}-M_{2}}{SD_{pooled}}$$
(2)$$SD_{pooled}=\sqrt{\frac{SS_{1}+SS_{2}}{df_{1}+df_{2}}}$$
where $M_{1}$ and $M_{2}$ represent the mean value of TME-related signature scores in group 1 and 2, respectively. $SD_{pooled}$ is the pooled standard deviation. $SS_{1}$ and $SS_{2}$ represent the sum of squares of deviation from mean in group 1 and 2, and $df_{1}$ and $df_{2}$ represent the degrees of freedom, respectively. In general, effect sizes below 0.2 are considered as small effect, those between 0.5 and 0.8 as medium effect, and those above 0.8 as large effect sizes [29].

### 2.4. Identification of Subtype-Specific TME Cells

The xCell method integrates the advantages of gene set enrichment with deconvolution approaches to assess the abundance of TME cells, and it can assess the infiltration values of multiple immune and stromal cells. Thus, we applied xCell to the breast cancer gene expression profile to estimate the infiltration abundance of 64 TME cells (including 34 immune cells and 30 stromal cells). This process was performed in R with R package “xCell”.

As the cell infiltration abundance varies slightly in a patient cohort, we developed a cell-based subtype set enrichment analysis (SubSEA) method to identify subtype-specific TME cells. We created a sample list L = <s_1_, s_2_…s_N_> by ranking the N samples in the dataset based on decreasing cell infiltration abundance for each cell. The samples in a certain subtype were mapped to the sample list L. The cell will be related to the specified subtype if the samples in the subtype are significantly enriched at the top or bottom of the entire ranked list L. We calculated a sample enrichment score (SES) using the weighted Kolmogorov–Smirnov statistic, which represents the extent to which the samples in a subtype are overrepresented toward top or bottom of the sample list L. The SES is calculated by walking down the list L, increasing a running sum statistic when we encounter a sample in the subtype, and decreasing it when we encounter a sample not in the subtype. This statistic has been used in gene set enrichment analysis (GSEA) previously [30], and it was used as a statistical test of sample sets of specific subtypes in the present study. At a given position *i* in the list L, we evaluated the fraction of samples in the subtype $F_{hit}\left(S,i\right)$ weighted by their cell infiltration abundance, and the fraction of samples not in the subtype $F_{miss}\left(S,i\right)$, as follows:(3)$$F_{hit}\left(S,i\right)=\underset{\begin{array}{c} s_{j}\in S\\ j\leq 1 \end{array}}{\sum }\frac{{\left|r_{j}\right|}^{P}}{N_{R}},\;where\;N_{R}=\underset{s_{j}\in S}{\sum }{\left|r_{j}\right|}^{P}$$
(4)$$F_{miss}\left(S,i\right)=\underset{\begin{array}{c} s_{j}\notin S\\ j\leq 1 \end{array}}{\sum }\frac{1}{N_{NotS}}$$
where $r_{j}$ is the cell infiltration abundance of sample *j*; $N_{NotS}$ represents the number of samples in the list L not in the subtype. With the position *i* walking down the list L, the SES is calculated as the maximum deviation from zero of $F_{hit}\left(S,i\right)$ − $F_{miss}\left(S,i\right).$ If the samples in a given subtype enrich near the top or bottom of the list, the SES will be large, however, if the samples are randomly distributed in the list, the SES will be small. The parameter *p* is used to weight the cell infiltration abundance of the samples in the subtype, and we set *p* = 1 in the study.

We used a gene-based permutation test procedure that preserved the sample labels and gene expression data to determine the statistical significance (nominal *p*-value) of the SES. Specifically, we permuted gene labels of gene expression profiles and recomputed cell infiltration abundance profiles and SES for the permutated data. When the observed SES > 0, the *p*-value was computed as *p*-value = M/N, where M is the number of permuted SESs greater than the observed SES, and N is the permutation times; when the observed SES < 0, *p*-value = M/N, where M is the number of permuted SESs less than the observed SES. Finally, we defined cells with *p*-value < 0.001 as subtype-specific.

### 2.5. Prognostic Analysis of Subtype-Specific TME Cells

In each subtype, we performed the univariate Cox proportional hazards regression analysis on the infiltration abundance of subtype-specific cells. For each statistically significant cell, the patients were separated into high-risk and low-risk subgroups on the basis of optimal cutoff of infiltration abundance, which was calculated from the R “survminer” package. Then, Kaplan–Meier analysis was performed to assess two subgroups’ survival distributions, and the log rank test was utilized to estimate the statistical significance.

### 2.6. Subtype-Specific Genomic Aberration

Somatic mutation profiles were transformed into a matrix composed of 0 and 1 by the R “SMDIC” package [31]. In the matrix, 1 corresponds to a gene being mutated in a sample, and 0 corresponds to the gene being wild type. Then, we retained genes with mutation frequencies greater than 1% across all samples in our analysis. To identify the subtype-specific mutation, the chi-square test was applied to genes to compare gene mutation status among different subtypes. Genes with FDR < 0.01 were defined as subtype-specific mutation genes. Similarly, we analyzed the variation in the driver CNV genes in breast cancer among different subtypes. We used a chi-square test to compare whether these genes experienced amplification or deletion with subtype specificity, respectively (FDR < 0.01).

### 2.7. Statistical Analysis

The two-sided Mann–Whitney U test is exploited to examine the statistical significance of continuous variables between each pair of subtype groups, such as immune score, stromal score, MHC score, T-cell-inflamed GEP score, and tumor mutation burden (TMB). The chi-square test is utilized to compare unordered categorical variables among subtypes, and the overall difference across all subtype groups was evaluated by Kruskal–Wallis test. The associations between TME cells and overall survival (OS) were examined using univariate Cox proportional hazards models. The Kaplan–Meier method was used to conduct the survival analysis, and for comparing the survival curves, log rank tests were utilized. In all the tests, unless otherwise stated, *p*-value <0.05 was considered significant. FDR-adjusted *p*-values were obtained by the Benjamini–Hochberg method. R software was used to conduct all the analyses (version 4.1.3, http://www.R-project.org (accessed on 1 March 2022)).

## 3. Results

### 3.1. Associations of TME-Related Features with Breast Cancer Subtypes

#### 3.1.1. Analysis of TME-Related Gene Set Signatures

To evaluate whether breast cancer molecular subtypes display distinct TME patterns, we gathered some TME-related gene set signatures from earlier studies that captured various biological aspects or cellular components, including immune response, vascularization, stroma compartment, and metabolic processes (see Materials and Methods). In the METABRIC dataset, we first calculated the effect size of expression levels of these signatures (ssGSEA scores) between samples of a specific subtype and other subtypes. The statistical significance of the differential expression levels of signatures was performed with the Mann–Whitney U test.

As illustrated in Figure 1A, Basal-like and HER2-enriched subtypes exhibited similar TME patterns characterized by higher expression levels in metabolism, vascularization, and immune signatures, together with low stromal signature expression levels. Luminal A subtype was enriched with high stromal expression levels, but low immune and metabolic expression levels. The Luminal B subtype was mainly associated with high levels of metabolism processes and low levels of vascularization and immune signatures, whereas the Normal-like subtype exhibited high expression in all signatures except metabolism processes. Cell metabolism is linked to human immunity, which has been described as playing a crucial role in controlling the function and activity of immune cells [32,33], indicating the potential mechanism for heterogeneous immune activity.

Taken together, our findings highlight the presence of a specific TME pattern characterizing each of the breast cancer molecular subtypes. The distinct TME-related gene signature activities led us to further explore the heterogeneity of the tumor immune microenvironment.

#### 3.1.2. Analysis of Immune-Related Feature Scores

We further investigated the differences in immune-related feature scores between each pair of breast cancer subtypes, including immune score, stromal score, tumor purity, MHC score, T-cell-inflamed GEP score, and TMB. Overall, all six characteristics were significantly correlated with the cancer subtypes (Kruskal–Wallis test *p* < 0.001; Figure 1B–G). Immune scores did not show a significant difference among Basal-like, HER2-enriched, and Normal-like subtypes. Both Luminal A and Luminal B subtypes were discovered to exhibit significantly lower immune scores than the other three subtypes (Figure 1B). The Normal-like subtype showed the highest stromal scores, while Basal-like and Luminal B subtypes had much lower scores than the other subtypes (Figure 1C). Tumor purity was highest in Luminal B subtype due to the lower stromal and immune scores (Figure 1D). MHC molecules use a variety of metabolic processes to present antigens to the immune system [34]. The Basal-like subtype was characterized by significantly higher MHC scores (Figure 1E), which means possibly stronger antigen presentation capacity. The T-cell-inflamed GEP score, related to antigen presentation, was predominantly associated with Basal-like and HER2-enriched subtypes. TMB was highest in the HER2-enriched subtype and lowest in the Normal-like subtype (Figure 1G).

From the above results, it can be summarized that Basal-like and HER2-enriched subtypes showed higher levels in terms of MHC score, T-cell-inflamed GEP score, and TMB, indicating strong immune infiltration and immunogenicity. Alternatively, the lower MHC score, T-cell-inflamed GEP score, or TMB in the other three subtypes may be due to the potential immune escape mechanisms of these subtypes. Furthermore, these specific features may help us to understand the immune microenvironment of breast cancer, and may provide a potential reference for immunotherapy of tumors.

#### 3.1.3. Analysis of Immune Target Genes

We next collected some important immune targets, including immunostimulatory and immunosuppressive genes, and some other immune-response-related receptors and ligands [9], and analyzed their expression differences between a specific subtype with the other subtypes by calculating the effect size.

Luminal A and Luminal B subtypes showed significantly lower expression levels in the majority of the examined immunological targets compared with the other three subtypes, which indicates that they were an immune-negative TME (Figure 1H). Notably, except for two immune inhibitory receptors, CD39 and interferon *α*/*β* receptor subunit 1 (IFNAR1), most of the evaluated immune targets did not show any evidence of immune reactivity in the Luminal A subtype, and CD39 and CD73 had higher expression levels in the Normal-like subtype. Adenosine A2a receptor (ADORA2A) and CD73 were highly expressed in the HER2-enriched subtype. As previously described, in the metabolic pathway, adenosine signaling has gradually become a critical regulator of tumor immunity. Combination immunotherapy techniques that include inhibitors of the hypoxia–CD39–CD73–A2a receptor pathway have been shown to improve cancer patient clinical outcomes [35]. However, the Basal-like subtype may be unsuitable for the pathway inhibition strategy due to its the negative association with ADORA2A and CD39. CD47 was an immunosuppressive target associated with Basal-like and Normal-like subtypes. CD47 is a cellular receptor with immunoregulatory properties that is widely expressed [36]. Furthermore, CD47–SIRPα has proven to release a self-protection signal leading to tumor immune escape, indicating that this pathway may be an effective target for ICB therapy [37].

In conclusion, it can be observed that immune targets are differentially expressed across breast cancer subtypes, and these conclusions may potentially guide the selection of immunotherapy regimens for breast cancer.

### 3.2. The Specific TME Cell Composition of Breast Cancer Subtypes

To acquire a comprehensive understanding of the microenvironment component in distinct subtypes of breast cancer, we used the xCell method to estimate the infiltrate abundance of 64 TME cells, and compared them between subtypes. We retained and visualized cells that differed significantly in at least one subtype (Figure 2A,B).

Overall, Basal-like and HER2-enriched subtypes were primarily composed of immune cells compared with the other subtypes. The Luminal A subtype had a low infiltration of immune cells, but higher abundances of stromal cells such as hematopoietic stem cells (HSCs). HSCs are rare, multipotent cells that can generate all the blood and immune system cells. Recent studies suggest that HSCs respond to infections and inflammatory signals in a direct and immediate manner [38]. Compared with the other subtypes, TME cells have a relatively balanced distribution in the Luminal B subtype, but the infiltration of both kinds of cells is much lower. Erythrocytes have higher infiltration only in the Luminal B subtype. A recent study showed that certain bacterial pathogens are spontaneously captured by erythrocytes in the circulation and presented to antigen-presenting cells (APCs) in the spleen [39]. Moreover, in the Normal-like subtype, the proportion of stromal cells is slightly higher than that of immune cells. We observed that dendritic cells (DCs) tended to have higher levels of infiltration in the Normal-Like subtype. Dendritic cells are the most efficient antigen-presenting cells [40]. Dendritic cells are the focus of tumor immunotherapy because of their function in producing protective adaptive immunity, and they are now becoming recognized as important regulators of the immune response within tumors [41].

Collectively, our findings demonstrate the significant heterogeneity in the cellular composition of the TME within each breast cancer subtype.

### 3.3. Analysis of Subtype-Specific Cells

We further used the cell-based subtype set enrichment analysis method to screen the subtype-specific infiltrated cells. With a *p*-value < 0.001, several subtype-specific cells were identified (Table 1 and Figure 3A). Specifically, Basal-Like and HER2-enriched subtypes were infiltrated by immune cells, such as CD4 Tcm, CD8 Tem, memory B cells, pro B cells, etc. Moreover, Luminal A, Luminal B, and Normal-like subtypes were mainly enriched with stroma cells. For example, the Luminal A subtype was specifically infiltrated by adipocytes, common myeloid progenitor cells (CMPs), and neutrophils; the Luminal B subtype was infiltrated by erythrocytes and granulocyte–macrophage progenitor cells (GMPs); and the Normal-like subtype was infiltrated by fibroblasts (Table 1 and Figure 3A).

Next, we applied the univariate proportional hazards model to test whether the subtype-specific TME cells are associated with patient prognosis in each subtype patient cohort. With *p*-values < 0.01, no subtype-specific cells were associated with prognosis in the Basal-like subtype or Normal-like subtype (Figure 3B,C). The bubble chart shows cells selected from the other three subtypes (Figure 3D–F). In the HER2-enriched subtype, the HSCs were associated with poor survival (HR > 1, *p* < 0.01) (Figure 3D), and four cells (memory B cells, CD4 naive T cells, and NK cells) were associated with good survival. In Luminal A, ten cells (CMPs, cDCs, iDCs, etc.) were significantly associated with improved outcomes (HR < 1, *p* < 0.01) (Figure 3E), and three cells (Th1 cells, osteoblasts, and Tregs) were associated with poor survival. Finally, four cells (mesangial cells, sebocytes, etc.) were significantly connected to poor survival in the Luminal B subtype (HR > 1, *p* < 0.01), whereas GMPs, smooth muscle cells, etc., were significantly connected to good outcomes in the Luminal B subtype (Figure 3F).

Furthermore, we performed the Kaplan–Meier analysis to investigate whether the above cells could predict OS in their corresponding subtypes. Interestingly, in the HER2-enriched subtype, high B cell infiltration corresponded to longer OS (log rank *p* < 0.001, Figure 4A), and natural killer T (NKT) cells were associated with improved OS (log rank *p* < 0.001, respectively, Figure 4B). Moreover, in the Luminal A subtype, high CD4(+) T effector memory cell (Tem) infiltration had better prognostic effects than lower infiltration (log rank *p* < 0.001, Figure 4C). Finally, we found that high infiltration of CD4(+) T central memory (Tcm) cells was correlated with improved OS in the Luminal B subtype (log rank *p* < 0.001, Figure 4D). The prognostic results of these specific cells in each subtype are demonstrated in the independent validation set (Figure 4E–H). The results indicate that high infiltration of these cells in the validation set is significantly associated with a good prognosis of the patients, although the statistical significance of some subtypes was weaker than in the training set. This situation was caused by the relatively small number of the samples in each subtype in the validation set. This suggests that cells in specific subtypes may have specific biological functions and prognostic significance. Moreover, the boxplot exhibits a marked difference in cell infiltration among subtypes (Figure 4I–L), and the subtype-specific prognostic cells have a high infiltrating level in the corresponding subtypes.

### 3.4. Breast Cancer Molecular Subtypes Display Distinct Genomic Aberration

As the heterogeneity of TME features across different breast cancer subtypes may be driven by specific genomic aberration, we first analyzed the mutational profiles of breast cancer for mining possible subtype-specific mutations. With the gene mutation frequency >1%, 116 genes were used for further analysis. After the chi-square test, 15 mutated genes, including PIK3CA, TP53, MUC16, GATA3, SYNE1, MAP3K1, DNAH11, CDH1, AKAP9, CBFB, RB1, CTCF, SMAD4, GPS2, and JAK1, showed significant differences (FDR < 0.01) among the breast cancer subtypes (Figure 5A).

The most commonly mutated genes in the METABRIC cohort were PIK3CA (49%), followed by TP53 (37%) and MUC16 (20%). PIK3CA showed a high mutation rate in the Luminal A subtype and a low mutation rate in the Basal-like subtype (Figure 5B). In breast cancer, immune response was lower in PIK3CA-mutated tumors compared with wild type tumors [42]. This result indicates that PIK3CA-mutated tumors may inhibit the immune response of Luminal A, which partly explains why Luminal A showed weak immune features. The TP53 mutations were majorly enriched in Basal-like and HER2-enriched subtypes compared with the others (91% and 73%); however, we observed that TP53 mutation rates were much lower (13%) within the Luminal A subtype. Breast cancer patients with TP53 mutations have been found to have higher immune infiltration levels [43], which further illustrates the positive immune landscape for Basal-like and HER2-enriched subtypes.

To comprehensively explore the genomic abnormalities in breast cancer, we further analyzed the CNA rates of 31 known copy number driver genes [44] in different molecular subtypes of breast cancer (Figure 6). The top five genes with the highest frequency of CNV across all breast cancer samples were CDH1 (63%), TP53 (55%), MAP2K4 (54%), NCOR1 (53%), and MYC (52%). The first four genes mainly experienced deletion variants and MYC mainly experienced amplification variants, and the amplification or deletion of these genes was specific among different subtypes (FDR < 0.01). CDH1 had the highest frequency of deletion in the Luminal A subtype compared to other subtypes, and TP53, MAP2K4, and NCOR1 were more likely to occur in the HER2-enriched subtype with copy number deletion. The Basal-like subtype was significantly associated with higher frequency amplification levels for MYC.

Therefore, the presence of these genomic abnormalities highlights the heterogeneous TME-related features and immune landscape among subtypes. Moreover, mutations in some genes can also provide novel insights for targeted therapy.

## 4. Discussion

Breast cancer can be mainly classified into Luminal A, Luminal B, HER2-enriched, Basal-like, and Normal-like subtypes according to the expression levels of 50 genes (PAM50). However, little is known regarding their heterogeneity in the TME-related features. A comprehensive analysis of TME-related feature differences across breast cancer subtypes is urgently needed. Here, we integrated multi-omics data of breast cancer to highlight the subtype-specific TME-related features, including TME-related gene set signatures, immune-related feature scores, immune target genes, and TME-related cell infiltration. We expect that our analysis may provide a landscape of TME-related features for different breast cancer types, and set the ground for the rationale tailoring of immunotherapy in breast cancer patients.

In the study, we first analyzed the expression of TME-related gene set signatures and suggested the specific TME pattern in different molecular subtypes. Increased research suggests that metabolism in cancer is important not just for tumorigenesis and survival, but also for the control of the antitumor immune response [45]. The metabolic pathways associated with TME were found to be activated during our investigation in the Basal-like, HER2-enriched, and Luminal B molecular subtypes (Figure 1A), revealing a complex relationship between cancer metabolism progression and TME. Thus, in these cancer subtypes, targeting metabolic pathways appears to be a feasible anticancer therapy method (Table 2). We next analyzed immune-related feature scores (immune score, stromal score, tumor purity, MHC score, T-cell-inflamed GEP score, and TMB), immune target genes, and TME-related cell infiltration (Figure 1, Figure 2 and Figure 3). Our analysis shows that Basal-like and HER2-enriched subtypes are linked to high immune scores, expression of most immune regulatory targets, and immune cell infiltration, suggesting that these subtypes could be defined as “immune hot” tumors. The GEP was related to antigen presentation and adaptive immune resistance. The higher GEP in Basal-like and HER2-enriched subtypes suggests that they are more suitable for immune therapy, such as immune checkpoint blockade (ICB) therapy. Luminal A and Luminal B subtypes were linked to low immune scores, expression of most immune regulatory targets, and immune cell infiltration, suggesting that these subtypes could be defined as “immune cold” tumors. However, the Luminal A tumors showed high expression levels of two immunosuppressed targets, namely, CD39 and IFNAR1. This means the Luminal A subtype may be suitable for CD39 and IFNAR1 inhibitor therapy. The Luminal B subtype exhibited a high expression level in terms of metabolism, suggesting that it is a potential candidate for anti-metabolite therapy. Moreover, the Normal-like subtype has relatively high immune and stromal features compared with Luminal A and Luminal B. These results suggest that the Normal-like subtype may be suitable for more diverse treatment strategies, such as ICB, or for targeting the adenosine pathway (Table 2). Based on our findings, we believe that identifying these TME traits will enable us to select the most effective immunotherapeutic strategy for specific subtypes in order to achieve the optimal therapeutic impact and clinical benefit.

Our results suggest that breast cancer molecular subtypes are highly heterogeneous, especially in terms of immunity; thus, identifying specific prognostic markers for breast cancer subtypes becomes particularly important. After performing survival analysis on the cells in each subtype, the subtype-specific prognostic cells were identified in HER2-enriched, Luminal A, and Luminal B subtypes. We have also verified our findings in the validation set. In the validation set, the number of samples in each subtype is smaller than that in the investigation set. This led to a substantial difference in the statistical significance of the survival curves, as well as the trends of the curves in the two sets. In the HER2-enriched subtype, patients with high infiltration of B cells and NKT cells had longer OS. High T helper 1 cell infiltration was linked to better prognosis in the Luminal A subtype. Within the Luminal B subtype, patients with high infiltration of CD4 central memory T cells had longer OS. These findings further illustrate the heterogeneity of the tumor microenvironment and the necessity for subtype-specific markers.

Furthermore, to elucidate the heterogeneity of the breast cancer TME at the multi-omics level, we assessed the mutational profiles and identified 15 subtype-specific mutated genes. These genes include some commonly mutated genes in breast cancer, such as PIK3CA, TP53, GATA3, and MAP3K15. PIK3CA is a gene frequently mutated in breast cancer [46], and its mutations have been a key target of therapeutic research in cancer, with the clinical trials of PI3K pathway inhibitors presently underway [47]. Mutations of TP53 are widespread in various cancer types, including breast cancer. The high frequency of TP53 mutations in breast tumors also enables the extensive application of p53-targeted therapies [48]. Patients with TP53 mutations and PIK3CA wild type tend to respond to immunotherapy, which means an improved clinical effect of ICI [49]. Of interest, the Basal-like subtype displayed a much higher mutation rate of TP53 and a much lower mutation rate of PIK3CA, which suggests the potential of ICIs for the Basal-like subtype. Furthermore, the somatic mutations in the MAPK signaling pathway suggest the possible sensitivity to MAPK inhibitors. Collectively, mutations in these genes have potential clinical significance, and combined with their subtype specificity, can inform the prognosis and/or treatment of specific patients (Table 2).

In conclusion, our study reveals that the breast cancer molecular subtypes have heterogeneous TMEs. This research provides us with a deeper understanding of the complicated connections between tumor cells and TME. Comprehensive analysis helps us to identify the prognostic marker and adopt the optimal treatment strategy for different subtypes.

## Figures and Tables

**Figure 1 genes-14-00044-f001:**
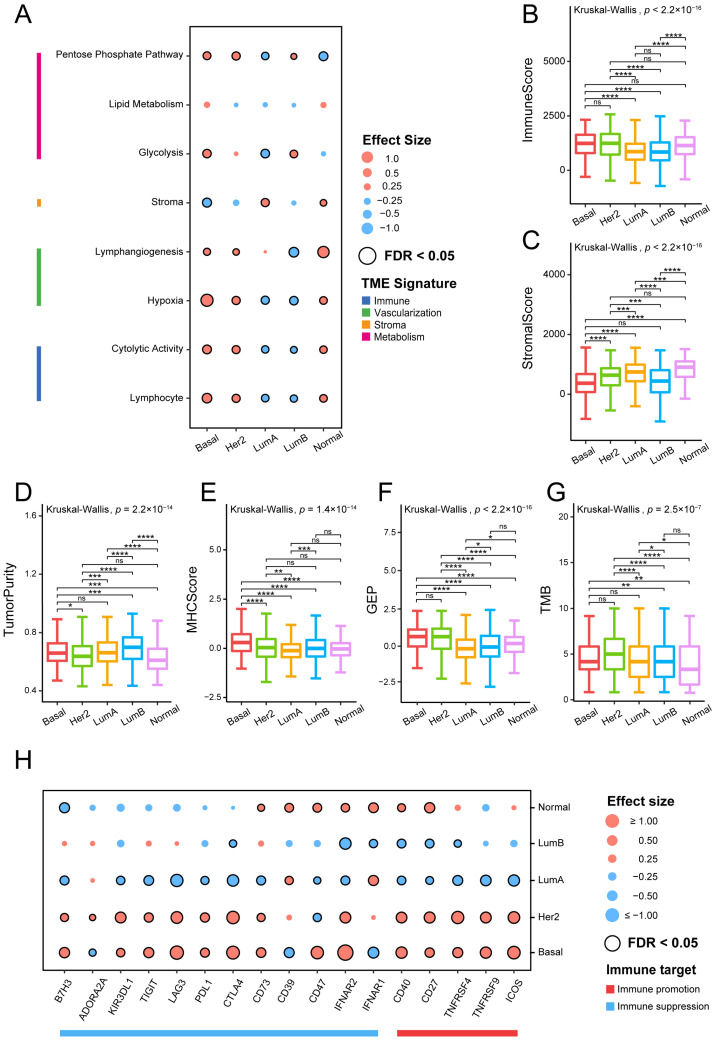
Association between TME-related signatures and breast cancer subtypes: (**A**) Associations between TME gene expression signatures and breast cancer subtypes. Effect size is a statistic used to reflect the different sizes between samples of a specific subtype and other subtypes, with higher and lower scores represented in red and green, respectively. The Mann–Whitney U test was used to calculate the statistical significance of differential gene expression signatures between a specific subtype and other subtypes. The statistically significant signatures in a specific subtype are marked with black circles (FDR < 0.05). Comparison of (**B**) immune score, (**C**) stromal score, (**D**) tumor purity, (**E**) MHC score, (**F**) GEP, and (**G**) TMB across all subtypes. The overall statistical significance (*p*-value) for each feature was calculated by Kruskal–Wallis test, and the statistical significance between each pair of subtype was calculated by Mann–Whitney U tests (****, *p* < 0.0001; ***, *p* < 0.001; **, 0.001 < *p* < 0.01; *, 0.01 < *p* < 0.05; ns, *p* > 0.05). (**H**) Association between immune target expressions and breast cancer subtypes.

**Figure 2 genes-14-00044-f002:**
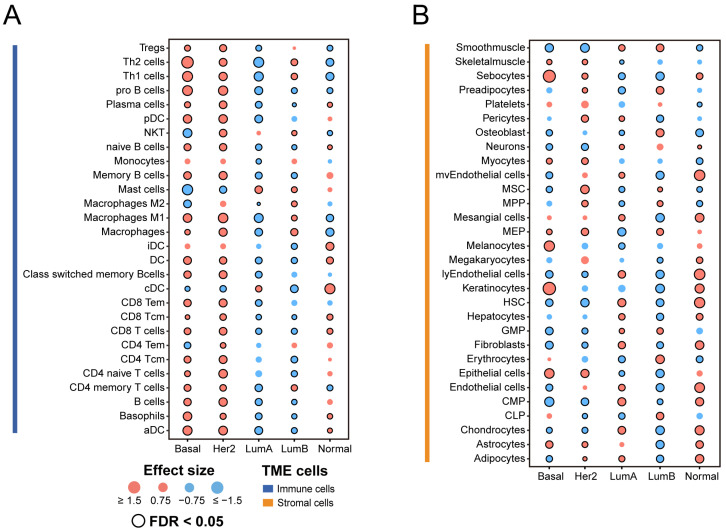
Association between TME cell infiltration levels and breast cancer subtypes: (**A**) Immune cell composition of breast cancer subtypes. (**B**) Stromal cell composition of breast cancer subtypes. Effect size is a statistic used to reflect the size of the difference between a specific subtype and other subtypes, with higher and lower scores represented in red and green, respectively. The Mann–Whitney U test was used to calculate the statistical significance of differential cell infiltration levels. The statistically significant results were marked with black circles (FDR < 0.05). Tregs, regulatory T cells; Th, helper T cells; NKT, natural killer T cell; DC, dendritic cell; aDC, activated dendritic cell; iDC, inactivated dendritic cell; pDC, plasmacytoid dendritic cell; cDC, classical dendritic cell; Tcm, central memory T cell; Tem, effector memory T cell.

**Figure 3 genes-14-00044-f003:**
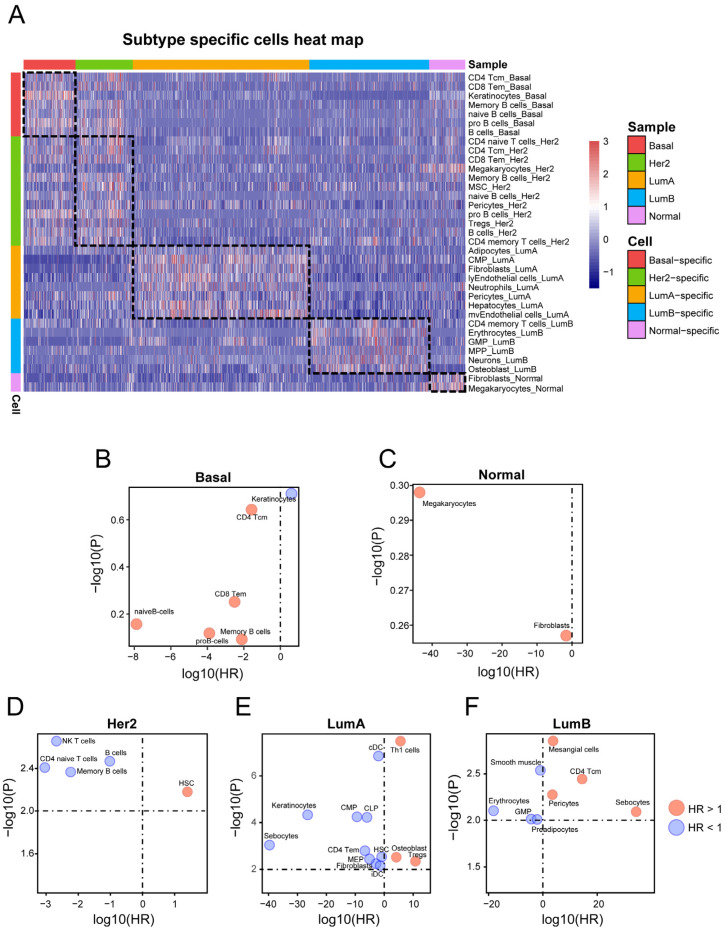
Identifying subtype-specific TME cells: (**A**) The heatmap represents the infiltration pattern of subtype-specific TME cells in each subtype. Each column represents a sample, and each color represents the subtype to which the sample belongs. (**B**–**F**) Associations between TME cells and OS using univariate Cox regression models in (**B**) Basal-like, (**C**) HER2-enriched, (**D**) Luminal A, (**E**) Luminal B, and (**F**) Normal-like subtypes, respectively. The x- and y-axes represent the log10 (hazard ratio) and the −log10 (*p*-value), respectively. The red bubbles represent HR > 1 and blue bubbles represent HR < 1.

**Figure 4 genes-14-00044-f004:**
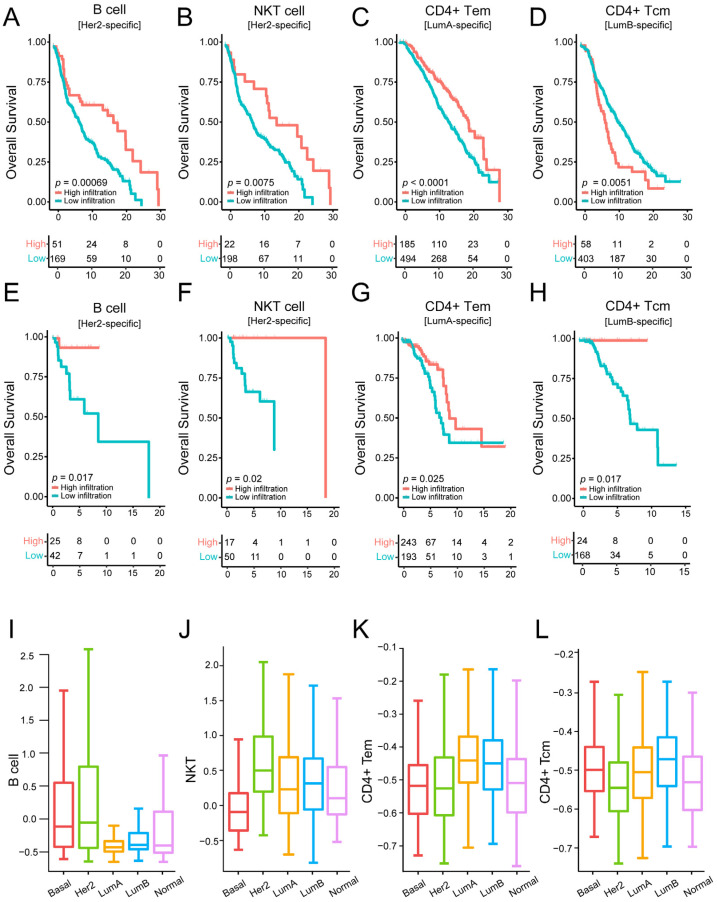
Identifying subtype-specific prognostic cells by performing survival analysis on the training dataset and validation dataset: (**A**–**D**) Kaplan–Meier survival analysis of OS comparing the high infiltration and low infiltration of B cells, NKT cells, CD4(+) Tem, and CD4(+) Tcm in HER2-enriched, Luminal A, and Luminal B subtypes from the METABRIC cohort (training dataset). (**E**–**H**) Kaplan–Meier survival analysis of OS comparing the high infiltration and low infiltration of B cells, NKT cells, CD4(+) Tem, and CD4(+) Tcm in HER2-enriched, Luminal A, Luminal B subtypes from the validation cohort (GDC TCGA cohort). (**I**–**L**) Comparison of infiltration levels of B cells, NKT cells, CD4(+) Tem, and CD4(+) Tcm across all subtypes.

**Figure 5 genes-14-00044-f005:**
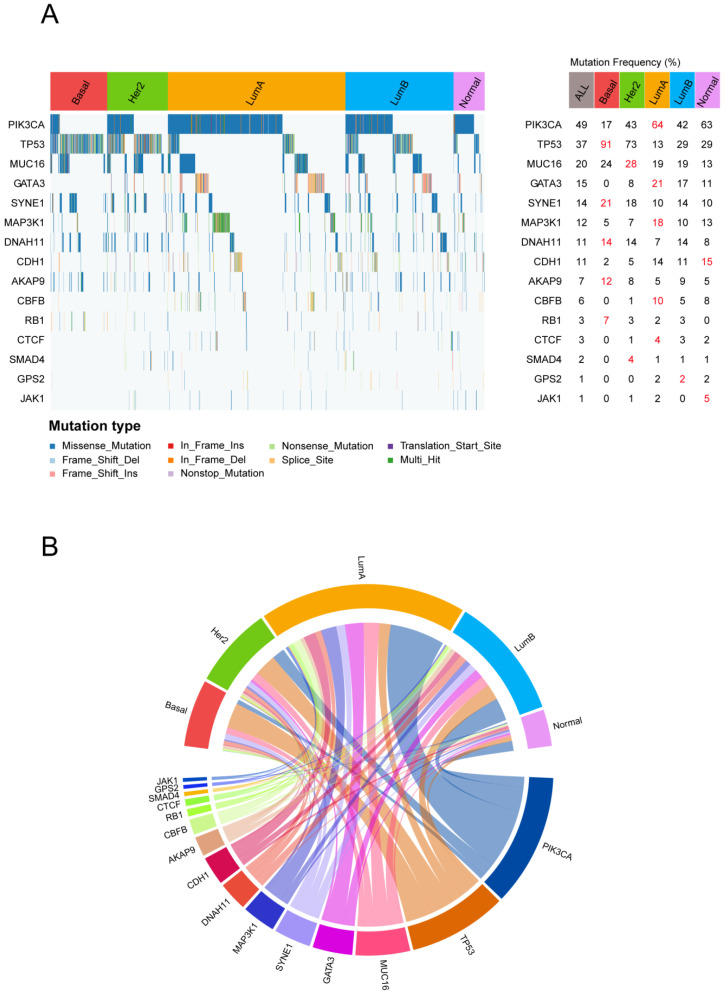
The specific gene mutation status for each subtype: (**A**) Waterfall plot of somatic mutation genes in different breast cancer subtypes (left). The statistical significance (*p*-value) was determined by the chi-square test, which was then adjusted by FDR. Genes with chi-square test FDR < 0.05 are shown. Gene mutation rate in each subtype (right). (**B**) The subtype-specific gene mutation patterns are exhibited in the circle plot. The top half of the circle represents the breast cancer molecular subtype, and the bottom half of the circle represents the mutation genes. If a gene is mutated in a sample of a subtype, the gene is connected to the subtype with a colored line.

**Figure 6 genes-14-00044-f006:**
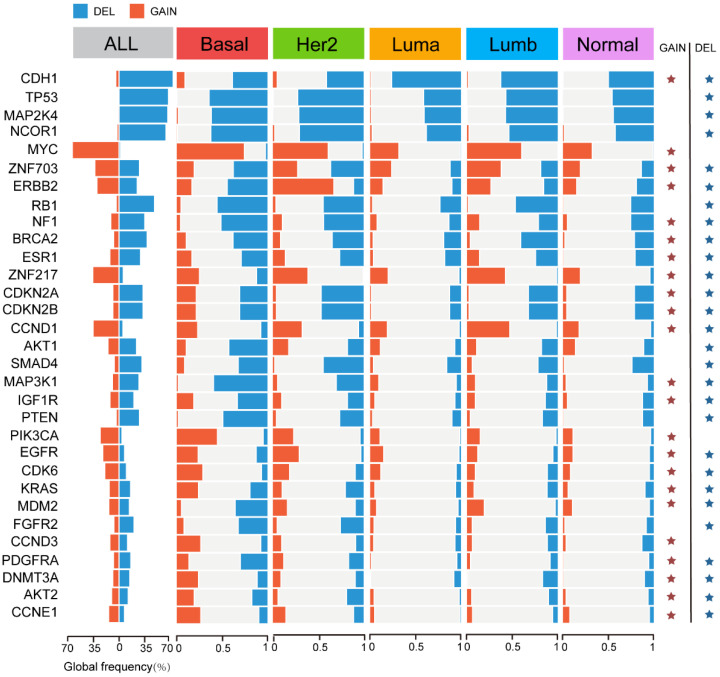
Copy number variation (CNV) within each breast cancer molecular subtype. CNV frequencies for the 31 breast cancer copy number driver genes across each molecular subtype. Significant differences (FDR < 0.01, chi-square test) are shown with an asterisk.

**Table 1 genes-14-00044-t001:** Subtype-specific cells identified in each molecular subtype with the cell-based subtype set enrichment analysis.

Subtype	Specific Cells
Basal-Like	CD4 Tcm, CD8 Tem, Keratinocytes, Memory B cells, naive B cells, pro B cells
HER2-enriched	CD4 naive T cells, CD4 Tcm, CD8 Tem, Megakaryocytes, Memory B cells, MSC, naive B cells, Pericytes, pro B cells, Tregs
Luminal A	Adipocytes, CMP, Endothelial cells, Neutrophils, Pericytes
Luminal B	CD4 memory T cells, Erythrocytes, GMP, MPP, Neurons, Osteoblasts
Normal-like	Fibroblasts, Megakaryocytes

**Table 2 genes-14-00044-t002:** Summary of TME features and therapy strategies in different subtypes.

Category	Basal	HER2	LumA	LumB	Normal-Like
Immune					
Immune-related signature	**+**	**+**	**−**	**−**	**+**
Immune score	**+**	**+**	**−**	**−**	**+**
Immune cell infiltration	**+**	**+**	**−**	**−**	**−**
Immune targets expression	**+**	**+**	**−**	**−**	**+**
GEP	**+**	**+**	**−**	**−**	**−**
Stromal					
Stromal score	**−**	**−**	**+**	**−**	**+**
Stromal cell infiltration	**−**	**−**	**+**	**−**	**+**
Vascularization					
Lymphangiogenesis	**+**	**+**		**−**	**+**
Hypoxia	**+**	**+**	**−**	**−**	**+**
Metabolism					
Pentose phosphate pathway	**+**	**+**	**−**	**+**	**−**
Glycolysis	**+**	**−**	**−**	**+**	
Subtype-specific mutation	TP53/SYNE1/DNAH11/AKAP9/RB1	MUC16/SMAD4	PIK3CA/GATA3/ MAP3K1/CBFB/ CTCF	GPS2	CDH1/JAK1
Therapy strategy	Immune checkpoint blockers (ICB)/ Anti-metabolite/ VEGF inhibitors	Immune checkpoint blockers (ICB)/ Anti- metabolite/ VEGF inhibitors/ hypoxia-CD39-CD73-A2aR pathway inhibitors	CD39 inhibitors/ IFNAR1 inhibitors	Anti- metabolite	VEGF inhibitors/ hypoxia-CD39-CD73-A2aR pathway inhibitors/ CD47 inhibitor/ IFNAR1 inhibitor

## Data Availability

Data are available from a public, open access repository. The datasets used and/or analyzed in the current study are available from the cBioPortal (https://www.cbioportal.org/ (accessed on 1 May 2022)) and the GDC TCGA data portal (https://portal.gdc.cancer.gov/ (accessed on 1 May 2022)).

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
