# Peer review of "Analysis of Tumor Microenvironment Heterogeneity among Breast Cancer Subtypes to Identify Subtype-Specific Signatures"

_genes, 2022, doi:10.3390/genes14010044_

Round 1

Reviewer 1 Report

The authors of the manuscript “Analysis of tumor microenvironment heterogeneity among breast cancer subtypes to identify subtype-specific signatures” aimed to uncover subtype specific tumor microenvironment (TME) features in the different molecular subtype of breast cancer. They used one publicly available breast cancer dataset (METABRIC) and explored mainly gene expression and mutational data.

The manuscript is well written, and the methodology is sound. However, the subject of this study is previously extensively studied, and the results are already known, as various papers have analyzed the TME of breast cancer in details. Also, the authors used only one cohort, but other breast cancer datasets are available. Overall, I could not suggest this paper for publication in Genes. I do have however a few questions and suggestions for the authors that could maybe improve the manuscript.

- I think that the authors could have given a few more details regarding the literature of the TME specifically in breast cancer in the Introduction.

- Line 80-81: please cite the paper from which the additional breast cancer dataset was taken from (https://doi.org/10.1002/emmm.201100801).

- Line 80-81: Why the authors used the above additional dataset? They could have used TCGA, that contains more samples, as a general validation cohort.

- Line 124-125: please cite the xCell method (https://doi.org/10.1186/s13059-017-1349-1)

- Section 3.3 paragraph 2: Could the authors explain why they found no TME cells associated with prognosis in the Basal-like and Normal-like subtypes?

- Figure 3B: The authors could add the Basal-like and Normal-like plots as well to show the non-significant results (could be as Supplementary data also).

- Section 3.4: The mutational landscape of breast cancer subtypes is already well-known. Thus, I believe this section is rather unnecessary and long. The authors could reduce the size of this section. Additionally, they could also show the landscape of copy number aberrations.

- Across the manuscript, some statements are a bit strong and should be rewritten. For example, in line 232-233 “these specific features bring us a more accurate view of immunotherapy for breast cancer”, or line 270-271 “these conclusions can help us to select appropriate immunotherapy regimens for patients more accurately”.

Reviewer 2 Report

Li et al. Analyzed tumor microenvironment heterogeneity among breast cancer subtypes and recognized subtype-specific signatures. The topic of this study is very interesting. The results supported the claimed conclusions. Manuscript is well written except for some minor typos.

1.The author screened the genes based on the mutation frequencies greater than 1% in the part “2.6. Subtype-specific mutation gene”. It's hard to understand whether it's 1 % across all samples or 1 % across each subtype.

2. In the introduction part, the author proposed some methods to estimate the cell infiltration. Why did xCell be chosen at last?

3. The author is required to give the full name of the section written in the first abbreviations display of the manuscript. For example, PAM50 needs to be changed to “Prediction Analysis of Microarray 50 (PAM50) in the introduction.

4. The author's description of “gene transcriptomic data” in Part 2.2 is not accurate. please use “transcriptomic data” or gene expression data

5. Does “SS1/2 in Part 2.3 represent the meaning of SS1 and SS2? If so, please change “SS1/2 to “SS1 and SS2 otherwise it will cause some ambiguity. If not, please specify what this parameter means. Meanwhile, please make the same correction for “df1/2””

6. In part 2.4 the phrase “be connected with  is not properly described and should be changed to “be related to”

7. There are some simple grammar errors in the manuscript. For example, in line 149 and 344,“cells” and “subgroup” may be “cell” and “subgroups”. There may be other such problems please find and correct.

Reviewer 3 Report

 N/A

Author Response

Thank you very much for your comments.

Reviewer 4 Report

Below are some suggestions to manuscript authors:

Line 35 - “...into Luminal A, Luminal B, HER2-enriched, Basal-like, and Normal-like (1) - Reference 1 is not a reference for the molecular classification of breast tumors, but a citation. https://doi.org/10.1038/35021093

Lines 57-58 “...TME cell infiltrated levels, such as CIBERSORT (8) - The reference 8 used is not about CIBERSORT, but rather a methodological citation - doi: 10.1007/978-1-4939-7493-1_12

Authors need to reorganize the sequence of references - the last one in the introduction was 10, and in sequence in materials and methods, they already start with 32-34.

Lines 354- 361 - Rearrange this paragraph. The authors need to provide a better rationale on the validation set with the significant discrepancy in the survival curves between the investigation and validation datasets, as well as better discuss these data in the discussion.

Round 2

Reviewer 1 Report

The authors have adequately addressed the questions and comments made by the reviewers and have performed additional analyses to strengthen their manuscript. I have no further questions. Congratulations for this work.